# On the Fundamental Limitations of Dual Static CVaR Decompositions in Markov Decision Processes

## Abstract

It was recently shown that dynamic programming (DP) methods for finding static CVaR-optimal policies in Markov Decision Processes (MDPs) can fail when based on the dual formulation, yet the root cause of this failure remains unclear. We expand on these findings by shifting focus from policy optimization to the seemingly simpler task of policy evaluation. We show that evaluating the static CVaR of a given policy can be framed as two distinct minimization problems. We introduce a set of "risk-assignment consistency constraints" that must be satisfied for their solutions to match and we demonstrate that an empty intersection of these constraints is the source of previously observed evaluation errors. Quantifying the evaluation error as the *CVaR evaluation gap*, we demonstrate that the issues observed when optimizing over the dual-based CVaR DP are explained by the returned policy having a non-zero CVaR evaluation gap. Finally, we leverage our proposed risk-assignment perspective to prove that the search for a single, uniformly optimal policy on the dual CVaR decomposition is fundamentally limited, identifying an MDP where no single policy can be optimal across all initial risk levels.

## 1 Introduction

The goal of reinforcement learning (RL) (Sutton & Barto, 2018) is to learn (sequential) decision-making policies such as to maximize some outcome (return) in a given environment, typically modeled as a Markov decision process (MDP). This is usually approached from the objective of maximizing the *expected* return, which has led to impressive successes in games (Silver et al., 2018; Vinyals et al., 2019) and content recommendation (Li et al., 2010). However, in safety-critical domains like healthcare, autonomous driving, or financial planning, some erroneous actions may lead to disastrous consequences. For instance, in the task of identifying the shortest path to an organ for surgery, paths at high risk of endangering the patient (e.g., as they are too close to an artery, a nerve, or a critical region of the brain), should be avoided (Baek et al., 2018). Automation in this context and other safety-critical domains therefore requires safe decision-making policies (Gottesman et al., 2019). This can be achieved by optimizing a *risk-averse* objective instead of simply maximizing the expected return (Artzner et al., 1999). In particular, conditional value-at-risk (CVaR), which is considered a gold standard risk measure in banking regulations (Basel Committee on Banking Supervision, 2019), has received a lot of focus in risk-averse RL (Prashanth et al., 2022).

More specifically, the *static* CVaR evaluation consists of computing the CVaR of a policy's cumulative random return. Unfortunately, optimizing a policy w.r.t. the static CVaR objective in MDPs has proven to be quite challenging since CVaR suffers from time inconsistency (Pflug & Pichler, 2016; Gagne & Dayan, 2021) and optimal policies may be history-dependent (Shapiro et al., 2014). To tackle these problems, prior work has considered dynamic programs (DPs) applied on augmented state spaces in both the primal and dual representations of risk measures. Working under the primal representation, states are augmented by keeping track of the running cumulative return (Boda et al., 2004; Bäuerle & Ott, 2011; Chow & Ghavamzadeh, 2014; Chow et al., 2018). However, such primal-based methods are considered practically inefficient since they require computing the value function on an unbounded continuous state space (Chow et al., 2015; Chapman et al., 2021; Li et al., 2022). The dual representation has therefore been identified as a promising direction, where the sequential decomposition of risk measures can be leveraged (Chow et al., 2015; Pflug

& Pichler, 2016). States here are augmented by keeping track of the current risk level (between 0 and 1). Although the resulting augmented state space is still continuous, discretization can be applied efficiently since risk levels are bounded (Chow et al., 2015; Li et al., 2022). Hence, Chow et al. (2015) proposed a Value Iteration (VI) procedure for CVaR in the dual representation, which served as the basis for many later developments in the field (Chow et al., 2015; Chapman et al., 2019; Stanko & Macek, 2019; Chapman et al., 2021; Rigter et al., 2021; Ding & Feinberg, 2022a;b), until Hau et al. (2023) showed counterexample MDPs where this procedure fails to recover the optimal policy. However, from these few empirical results alone, it is not possible to understand *what* makes CVaR VI fail, and therefore *how*, or even *if*, it can be fixed.

**Contributions**   The main goal of this work is to diagnose the root causes of recently observed failures in the static dual CVaR DP decomposition. To this end, we first establish a formal analysis framework that recasts the static CVaR evaluation and its DP decomposition as two distinct optimization problems over perturbations. Leveraging this perspective, we identify a set of *risk-assignment consistency constraints* that must be satisfied for the DP evaluation of a policy to be accurate. We show that the previously observed cases where the DP decomposition returned a suboptimal policy are explained by these constraints being inconsistent for the returned policy. We then leverage this constraint-based view to present an MDP where the action constraints required for optimality at different initial risk levels are irreconcilable. This proves that no single risk-dependent policy can be *uniformly optimal*, revealing a fundamental limitation of the pursuit of a single, universally optimal policy for all risk levels via the dual decomposition, independent of the DP algorithm used. Practically, these findings suggest that the standard approach of training a single universal policy on the risk-augmented state space is structurally flawed. Our results indicate that practitioners should instead favor training specific policies for targeted risk profiles, akin to primal-based decomposition methods. To increase readability, we postpone most of the proofs to the appendix.

## 2    Static Risk-Averse Reinforcement Learning

Following Puterman (2014), let us define a finite Markov decision process (MDP) by a tuple $(\mathcal{S}, \mathcal{A}, P, \mathcal{R}, s_0, \gamma)$ where $\mathcal{S}$ is a finite state space, $\mathcal{A}$ is a finite action space, $P : \mathcal{S} \times \mathcal{A} \mapsto \Delta(\mathcal{S})$ is the transition function between states ($\Delta$ denoting the probability simplex), $\mathcal{R} : \mathcal{S} \times \mathcal{A} \times \mathcal{S} \mapsto [0, R_{\max}]$ is the reward function, $s_0 \in \mathcal{S}$ is the initial state from which the process begins, and $\gamma \in [0, 1)$ is a discount factor. At each time step $t \in \mathbb{N}_0$ of a trajectory, an agent performs an action $A_t \in \mathcal{A}$ in the current state $S_t \in \mathcal{S}$ according to some decision policy $\pi$. This leads to a transition into the state $S_{t+1}$ sampled from $P(S_t, A_t)$, following which the agent receives the reward $R_{t+1} = \mathcal{R}(S_t, A_t, S_{t+1})$. This process is repeated over an horizon of $T$ time steps. Throughout this paper, we adopt the convention of using uppercase letters to distinguish objects subject to random realization, such as referring to the state $S_t = s \in \mathcal{S}$. A table of notations can be found in Appendix A.

**Remark 1.** *Imposing a deterministic reward function and initial state is done with minimal loss of generality. Any MDP with a stochastic reward function or initial state can be converted to an equivalent MDP with deterministic counterparts, provided the distributions are discrete. This transformation involves augmenting the state space, with the initial action having no effect on the state transition (Sutton & Barto, 2018).*

**Assumptions 1.** *Throughout this work, we consider finite state and action spaces ($|\mathcal{S}| < \infty, |\mathcal{A}| < \infty$). We focus on the finite-horizon setting with horizon $T$ and we include a discount factor $\gamma \in [0, 1)$ to remain consistent with general formulations. Crucially, we restrict our attention to* deterministic *policies. This restriction is standard in the dual CVaR decomposition literature (Chow et al., 2015; Hau et al., 2023) to isolate external risk (stochasticity of the environment) from internal risk (policy randomization), simplifying the analysis of the risk level updates.*

**Policies**   We consider agent policies as deterministic action-selection mechanisms. The most general form of policies is *history-dependent* policies $\pi_h : \mathcal{H} \rightarrow A$, where $\mathcal{H}$ is the set of histories defined as all previous states and actions encountered prior to the current action-selection. Let $H_t := (S_0, A_0, S_1, A_1, S_2, \ldots, S_t)$ and $\mathcal{H}_t$ respectively denote the current history and the set of possible histories at time $t$. At time $t = 0$, the set of histories is limited to the initial state, that is $\mathcal{H}_0 := \{(s_0)\}$. For subsequent steps it is defined recursively as the combination of possible previous histories with the previous action and current state concatenated, that is $\mathcal{H}_{t+1} := \mathcal{H}_t \times \mathcal{A} \times \mathcal{S}$. Allowing a slight abuse of notation, we extend transition and action-selection

dynamics to histories and define the probability of observing a given history $H_t$ given policy $\pi_h$ as

$$P^{\pi_h}(H_t) := \prod_{\tau=0}^{t-1} P(S_{\tau+1}|S_\tau, A_\tau)\mathbb{1}[A_\tau = \pi_h(H_\tau)],$$

where $\mathbb{1}$ denotes the indicator function.

Because of the degree of complexity brought upon by policies operating on the possibly immense set of histories, we are often interested in Markovian policies $\pi : \mathcal{S} \to \mathcal{A}$, a special case of history-dependent policies where actions are selected only based on the current state. Hereafter, $\pi$ will be used to denote Markovian policies while $\pi_h$ will denote history-dependent policies.

**Standard objective** The return associated with a history $H$ is defined as the discounted sum of rewards

$$\mathcal{R}_{0:T}^H := \sum_{t=0}^{T-1} \gamma^t R_{t+1}, \tag{1}$$

where $R_{t+1} = \mathcal{R}(S_t, A_t, S_{t+1})$. Given that trajectories in an MDP are generated using a random process, due to state transitions being stochastic, we denote the random return of a trajectory generated by policy $\pi_h$ as a random variable $Z^{\pi_h}$, taking value $\mathcal{R}_{0:T}^H$ (Eq. 1) with $H \sim P^{\pi_h}$. The standard RL objective (Sutton & Barto, 2018) is to identify the optimal policy $\pi_h^\star$ that maximizes the *expected return* over histories

$$\pi_h^\star \in \arg\max_{\pi_h} \mathbb{E}[Z^{\pi_h}]. \tag{2}$$

It is known that Equation 2 can always be solved by a Markovian policy $\pi$ (Szepesvári, 2022). Unfortunately, because the expectation only weights random outcomes according to their likelihood without taking their value into account, the optimal policy according to this objective may lead to *catastrophic* outcomes over some trajectories (Mannor et al., 2007). In critical applications where such trajectories should be avoided, one may optimize a risk-averse objective instead, where large negative outcomes are assigned higher importance.

## 2.1 Static CVaR for risk aversion

Let $Z \in \mathbb{R}$ denote a bounded variable on a probability space $(\Omega, \mathcal{F}, \mathbb{P})$, with cumulative distribution function $F_Z(z) = \mathbb{P}[Z \leq z]$ for some threshold $z \in \mathbb{R}$. Denote the *Value-at-Risk* (VaR) at risk level $\alpha \in (0, 1]$ as $\text{VaR}_\alpha[Z] := \min\{z \mid F_Z(z) \geq \alpha\}$. Assuming that $Z$ represents a payoff that should be maximized, the *conditional-value-at-risk (CVaR)* (Rockafellar & Uryasev, 2000; Föllmer & Schied, 2016) at risk level $\alpha$ is given by

$$\text{CVaR}_\alpha[Z] := \frac{1}{\alpha} \int_0^\alpha \text{VaR}_\beta(Z)\mathrm{d}\beta = \underbrace{\inf_{\xi \in \Xi_\alpha(\mathbb{P})} \mathbb{E}_\xi[Z]}_{\text{dual formulation}}, \tag{3}$$

where $\Xi_\alpha(\mathbb{P}) := \left\{\xi : \omega \mapsto \left[0, \frac{1}{\alpha}\right] \,\middle|\, \int_{\omega \in \Omega} \xi(\omega)\mathbb{P}(\omega)d\omega = 1\right\}$ defines the $\text{CVaR}_\alpha$ *risk envelope* around distribution $\mathbb{P}$ and $\mathbb{E}_\xi[Z]$ is the $\xi$-reweighed expectation of $Z$. If $Z$ has a continuous distribution, it is well known that we have $\text{CVaR}_\alpha[Z] = \mathbb{E}[Z|Z \leq \text{VaR}_\alpha[Z]]$, which can be interpreted as the expected value of the worst $\alpha$ outcomes of $Z$. Note that CVaR is monotonically increasing in $\alpha$ with edge cases representing $\text{CVaR}_0[Z] = \text{ess}\inf[Z]$ and $\text{CVaR}_1[Z] = \mathbb{E}[Z]$.

The *dual formulation* in Equation 3 shows that the CVaR can be expressed as an optimization problem, where the objective is to find perturbations $\xi$ applied to the stochastic generative process of variable $Z$ such as to minimize its expectation. From the definition of the risk envelope $\Xi_\alpha(\mathbb{P})$, we can observe that the perturbations enjoy two interesting properties. First, because $\xi$ represents *multiplicative* interventions on an event's likelihood, it only affects events with nonzero probability. Also, because the largest magnitude of perturbations on an event is $\frac{1}{\alpha}$, one can view $\frac{1}{\alpha}$ as a *perturbation budget* which is naturally minimal at $\alpha = 1$ and increases as $\alpha$ decreases to 0, simultaneously recovering the monotonically increasing property of CVaR and its edge cases.

**CVaR-RL objective** Recalling that the random return of policy $Z^{\pi_h}$ is a random variable, one can therefore define the static CVaR of a policy as

$$\text{CVaR}_\alpha\left[Z^{\pi_h}\right] := \min_{\xi \in \Xi_\alpha(P^{\pi_h})} \sum_{H \in \mathcal{H}_T} P^{\pi_h}(H)\xi(H)\mathcal{R}_{0:T}^H, \tag{4}$$

where we shall hereafter refer to $\xi$ as *history perturbations*. By emphasizing negative outcomes, Equation 4 naturally yields the CVaR-RL risk-averse objective (Tamar et al., 2015)

$$\pi_h^\star \in \arg\max_{\pi_h} \text{CVaR}_\alpha\left[Z^{\pi_h}\right], \tag{5}$$

where one aims to instead find a policy maximizing the $\text{CVaR}_\alpha$ of its random return. Because $\text{CVaR}_\alpha[Z^{\pi_h}]$ can be intuitively interpreted as the expectation of the worst $\alpha$ trajectories when following policy $\pi_h$, optimizing the CVaR-RL objective should yield policies less prone to catastrophic outcomes than the standard RL objective (Eq. 2), with a lower $\alpha$ leading to increased cautiousness.

## 2.2 CVaR-RL dynamic decomposition

Trajectory-level computation of static CVaR (Eq. 4) is impractical because it requires computing $P^{\pi_h}$ for all trajectories, which can be prohibitive for large state and action spaces. Fortunately, the CVaR decomposition Theorem (Chow et al., 2015; Pflug & Pichler, 2016) grants a recipe for expressing the evaluation at state-level.

**Theorem 1** (CVaR decomposition, Thm. 2 from Chow et al. (2015)). *For any time step $t \geq 0$, denote by $Z_{t:T}^{\pi_h}$ the return from time $t + 1$ onward under history-dependent policy $\pi_h$. Given current history $H_t$, the $CVaR_\alpha$ of $Z_{t:T}^{\pi_h}$ obeys the following decomposition:*

$$\text{CVaR}_\alpha\left[Z_{t:T}^{\pi_h} \mid H_t\right] = \min_{\tilde{\xi} \in \Xi_\alpha(P(\cdot|S_t,A_t))} \sum_{s' \in \mathcal{S}} P(s'|S_t,A_t)\tilde{\xi}(s')\,\text{CVaR}_{\alpha \cdot \tilde{\xi}(s')}\left[Z_{t:T}^{\pi_h} \mid H'\right],$$

*where $\tilde{\xi}$ are perturbations over next state transitions, action $A_t$ is given by policy $\pi_h(H_t)$, and $H' = H_t \cup (A_t, s')$ is a possible history realization at time $t + 1$.*

**Remark 2.** *We distinguish perturbations over next states ($\tilde{\xi}$) from perturbations over histories ($\xi$). While both perturbations impact the sampling of events, $\tilde{\xi}$ also updates the ongoing risk-level as dictated by the CVaR decomposition theorem (Thm. 1). When accumulated over an entire history, state perturbations implicitly yield history-level perturbations, but the connection between the two perturbation levels is complex and will be a core component of our analysis.*

**Risk-dependent policies** Theorem 1 shows that the $\text{CVaR}_\alpha$ at any given time $t$ can be expressed as combination of $\text{CVaR}_{\alpha'}$ values of possible next states $s'$ for updated risk levels $\alpha' = \alpha \cdot \tilde{\xi}(s')$ at time $t + 1$. The running risk level therefore contains all the information necessary to compute the CVaR of a history-dependent policy $\pi_h$. This motivated Chow et al. (2015) to introduce the *risk-augmented* state space $\tilde{\mathcal{S}} : \mathcal{S} \times (0,1]$, defined for any state $s \in \mathcal{S}$ and risk level $y \in (0,1]$, and the corresponding *risk-dependent* Markovian policies on the augmented state space $\tilde{\pi} : \mathcal{S} \times (0,1] \to \mathcal{A}$. Chow et al. (2015) suggested that operating over $\tilde{\mathcal{S}}$ would suffice to retrieve the optimal history-dependent policy and evaluate its corresponding static CVaR through a *value function* mimicking the mechanism of Theorem 1.

**Definition 1** (Risk-dependent-policy value function). *The value function $\boldsymbol{v}^{\tilde{\pi}}(s,y)$ of any risk-dependent policy $\tilde{\pi}$ is the solution to*

$$\boldsymbol{v}_{t+1}^{\tilde{\pi}}(s,y) = \min_{\tilde{\xi} \in \Xi_y(P(\cdot|s,a))} \sum_{s' \in \mathcal{S}} P(s'|s,a)\tilde{\xi}(s') \left[\mathcal{R}(s,a,s') + \gamma \boldsymbol{v}_t^{\tilde{\pi}}(s',y')\right] \tag{6}$$

*where $a = \tilde{\pi}(s,y)$ is the action selected by the policy, $y' = y \cdot \tilde{\xi}(s')$ is the subsequent risk level, and $\boldsymbol{v}_0^{\tilde{\pi}}(s,y) = 0$ for all states $s \in \mathcal{S}$ and risk levels $y \in (0,1]$. We let $\boldsymbol{v}^{\tilde{\pi}} := \boldsymbol{v}_T^{\tilde{\pi}}$.*

Crucially, any risk-dependent policy $\tilde{\pi}$ induces a corresponding history-dependent policy $\tilde{\pi}_h^\alpha$ for a given initial risk level $Y_0 = \alpha$. At time $t$, given the risk-augmented state $(S_t, Y_t)$ and the selected action $A_t = \tilde{\pi}(S_t, Y_t)$, the subsequent risk level is updated to $Y_{t+1} = Y_t \cdot \tilde{\xi}^\star(S_{t+1}|S_t, Y_t, A_t)$, where the optimal perturbations $\tilde{\xi}^\star$ denote the solution to the value function $\boldsymbol{v}^{\tilde{\pi}}(S_t, Y_t)$. Repeating this process $t$ times allows to compute the action $\tilde{\pi}_h^\alpha(H_t)$ for any history $H_t$. In the remainder of this paper, we slightly abuse terminology and refer to a risk-dependent policy's static CVaR to represent the static CVaR of its history-dependent counterpart.

In light of this correspondence between risk-dependent and history-dependent policies, Chow et al. (2015) proposed a Value Iteration algorithm, which we refer to as *CVaR VI*, to find the risk-dependent policy with the optimal value function $\tilde{\pi}^\star \in \arg\max_{\tilde{\pi}} \boldsymbol{v}^{\tilde{\pi}}(s_0, \alpha)$. Tentative proofs (Chow et al., 2015; Li et al., 2022) claimed $\tilde{\pi}^\star$ represented a risk-dependent version of the CVaR-optimal history-dependent policy $\pi_h^\star$, hence presenting a dynamic program decomposition of the CVaR-RL objective (Eq. 5). The optimality of the policy returned by CVaR VI was however refuted by Hau et al. (2023), who presented a counterexample MDP where the algorithm returns a suboptimal policy.

**CVaR evaluation gap**  In this work, we aim to explain *why* CVaR VI fails at a more fundamental level. To this end, we focus on the root cause, that is the discrepancy between the value function of a risk-dependent policy and its corresponding static CVaR, which we formally define as the *CVaR evaluation gap*

$$\boldsymbol{v}^{\tilde{\pi}}(s_0, \alpha) - \text{CVaR}_\alpha\left[Z^{\tilde{\pi}_h^\alpha}\right]. \tag{7}$$

A positive gap indicates that the value function of the risk-dependent policy overestimates its history-dependent counterpart's true CVaR.

## 3 An Explicit Mapping from the Value Function to the Static CVaR

To diagnose the source of the CVaR evaluation gap (Eq. 7), we first formalize the relationship between the value function of a risk-dependent policy (Eq. 6) at the initial state $(s_0, \alpha)$ and its corresponding static CVaR. We find that both problems can be cast as distinct, but closely related, perturbation optimization problems. Although they operate over different optimization spaces, respectively state-level perturbations $\tilde{\xi}$ and history-level perturbations $\xi$, we can derive a formal mapping from one problem to the other. Exploiting the mapping's properties, we then establish that the value function of a risk-dependent policy constitutes an upper bound on the static CVaR of the policy.

While the static CVaR evaluation (Eq. 4) is inherently defined as a minimization problem, evaluating the value function of a risk-dependent policy (Eq. 6) is defined as requiring $T$ recursive minimization problems, making for more cumbersome mathematical manipulations. In order to ease the manipulation of the latter, we first define the value function of a risk-dependent policy under a *fixed* set of *state-level perturbations*. For a fixed risk-dependent policy $\tilde{\pi}$, let $\tilde{\xi}$ denote a complete specification of state-level perturbations such that $\tilde{\xi}(\cdot|s, y, a) \in \Xi_y(P(\cdot|s, a))$ for all states $s \in \mathcal{S}$, risk levels $y \in (0, 1]$, and actions $a \in \mathcal{A}$. We further define $\tilde{\Xi} := \left\{\tilde{\xi} : \mathcal{S} \times (0, 1] \times \mathcal{A} \to \mathcal{S} \times \mathbb{R}^+ \,|\, \tilde{\xi}(\cdot|s, y, a) \in \Xi_y(P(\cdot|s, a)) \,\forall(s, y, a) \in \mathcal{S} \times (0, 1] \times \mathcal{A}\right\}$ as the set of all such valid state-level perturbations.

**Definition 2** (Policy-perturbations value function)**.** *The* policy-perturbations value function *of a risk-dependent policy $\tilde{\pi}$ under state-level perturbations $\tilde{\xi} \in \tilde{\Xi}$ is the solution to:*

$$\boldsymbol{v}_{t+1}^{\tilde{\pi}, \tilde{\xi}}(s, y) = \sum_{s' \in \mathcal{S}} P(s'|s, a) \, \tilde{\xi}(s'|s, y, a) \left[R(s, a, s') + \gamma \boldsymbol{v}_t^{\tilde{\pi}, \tilde{\xi}}(s', y')\right], \tag{8}$$

*where we used action $a = \tilde{\pi}(s, y)$, updated risk level $y' = y \cdot \tilde{\xi}(s'|s, y, a)$, and $\boldsymbol{v}_0^{\tilde{\pi}, \tilde{\xi}}(s, y) = 0$ for all states $s \in \mathcal{S}$ and risk levels $y \in (0, 1]$. We let $\boldsymbol{v}^{\tilde{\pi}, \tilde{\xi}}(s, y) := \boldsymbol{v}_T^{\tilde{\pi}, \tilde{\xi}}(s, y)$.*

This definition differs from the one in Equation 6 because it concerns *fixed* state-level perturbations $\tilde{\xi}$ instead of computing them recursively at every iteration. The value function of a risk-dependent policy $\tilde{\pi}$ can now be seen as finding the best possible perturbations $\tilde{\xi} \in \tilde{\Xi}$ to minimize this value.

**Lemma 1** (Value function evaluation). *Under the conditions of Assumptions 1, the value function evaluation of a risk-dependent policy $\tilde{\pi}$ (Eq. 6) is equivalent to solving*

$$\boldsymbol{v}^{\tilde{\pi}}(s, y) = \min_{\tilde{\boldsymbol{\xi}} \in \tilde{\boldsymbol{\Xi}}} \boldsymbol{v}^{\tilde{\pi}, \tilde{\boldsymbol{\xi}}}(s, y), \tag{9}$$

*where the above holds for all state-risk level pairs $(s, y)$ simultaneously, meaning a single state-level perturbations set $\tilde{\boldsymbol{\xi}}^{\star}$ is optimal for all $(s, y)$.*

We now have two distinct single-step optimization problems for static CVaR evaluation: the static evaluation over history perturbations $\xi$ (Eq. 4) and the value function evaluation over state-level perturbations $\tilde{\boldsymbol{\xi}}$ (Eq. 9). The two problems are in fact intimately connected. That is, any state-level perturbations $\tilde{\boldsymbol{\xi}}$ can be mapped to corresponding history-level perturbations $\xi$ by taking the product of state-level perturbations along each history. For an initial risk level $\alpha$ and history $H \in \mathcal{H}_T$, we define this mapping as

$$\zeta_{\alpha}^{\tilde{\boldsymbol{\xi}}}(H) \coloneqq \prod_{t=0}^{T-1} \tilde{\boldsymbol{\xi}}(S_{t+1} | S_t, Y_t, A_t),$$

where the risk levels $Y_t$ are incremented following $\tilde{\boldsymbol{\xi}}$ and starting from $Y_0 = \alpha$. We now show that the mapping $\zeta_{\alpha}^{\tilde{\boldsymbol{\xi}}}$ produces valid history perturbations recovering the value function for $\tilde{\pi}$ at the history-level.

**Proposition 1** (State-level perturbations evaluation correspondence). *Under the conditions of Assumptions 1, for any risk-dependent policy $\tilde{\pi}$, initial risk level $\alpha$, and state-level perturbations $\tilde{\boldsymbol{\xi}} \in \tilde{\boldsymbol{\Xi}}$, the mapping $\zeta_{\alpha}^{\tilde{\boldsymbol{\xi}}}$ produces valid history perturbations, that is $\zeta_{\alpha}^{\tilde{\boldsymbol{\xi}}} \in \Xi_{\alpha}\left(P^{\tilde{\pi}_h^{\alpha}}\right)$, for which we have*

$$\sum_{H \in \mathcal{H}_T} P^{\tilde{\pi}_h^{\alpha}}(H) \, \zeta_{\alpha}^{\tilde{\boldsymbol{\xi}}}(H) \, \mathcal{R}_{0:T}^H = \boldsymbol{v}^{\tilde{\pi}, \tilde{\boldsymbol{\xi}}}(s_0, \alpha).$$

Because Proposition 1 applies to *any* state-level perturbations set $\tilde{\boldsymbol{\xi}} \in \tilde{\boldsymbol{\Xi}}$, in particular it applies to the optimal state-level perturbations set $\tilde{\boldsymbol{\xi}}^{\star} \in \arg\min_{\tilde{\boldsymbol{\xi}} \in \tilde{\boldsymbol{\Xi}}} \boldsymbol{v}^{\tilde{\pi}, \tilde{\boldsymbol{\xi}}}(s_0, \alpha)$ for a fixed $\tilde{\pi}$. It follows that the value function evaluation (Lemma 9) is always an upper-bound to the true static CVaR of a policy (Eq. 4).

**Corollary 1** (Static CVaR upper-bound). *Under the conditions of Assumptions 1, for any risk-dependent policy $\tilde{\pi}$ and initial risk level $\alpha$, we have*

$$\mathrm{CVaR}_{\alpha}\left[Z^{\tilde{\pi}_h^{\alpha}}\right] \leq \boldsymbol{v}^{\tilde{\pi}}(s_0, \alpha).$$

As a result of Corollary 1, the CVaR evaluation gap (Eq. 7) is non-zero if and only if the optimal history perturbations $\xi^{\star}$ are not in the image of the mapping $\zeta_{\alpha}$. That is, if the best global (history-level) perturbations cannot be decomposed into a sequence of valid local (state-level) perturbations, the value function evaluation will return an erroneous estimation of the policy's static CVaR.

## 4 Characterizing the CVaR Evaluation Gap

We established that the value function of a risk-dependent policy provides an upper bound on its true static CVaR. In this section, we now investigate the exact conditions under which the upper bound is strict, leading to a CVaR evaluation gap. We show that the gap emerges when the optimal history-level perturbations are not *realizable* at the state level, a property that we formalize through a set of consistency constraints.

**Definition 3** (Realizable trajectory perturbations). *For a given risk-dependent policy $\tilde{\pi}$ and initial risk level $\alpha \in (0, 1]$, trajectory perturbations $\xi \in \Xi_{\alpha}\left(P^{\tilde{\pi}_h^{\alpha}}\right)$ are* realizable *if there exists state-level perturbations $\tilde{\boldsymbol{\xi}} \in \tilde{\boldsymbol{\Xi}}$ such that $\zeta_{\alpha}^{\tilde{\boldsymbol{\xi}}} = \xi$.*

The existence of such state-level perturbations $\tilde{\boldsymbol{\xi}}$ hinges on our ability to define a sequence of intermediate risk levels $Y_t$ that are mutually consistent for all histories. Recall that $\mathcal{H} \coloneqq \bigcup_{t=1}^{T} \mathcal{H}_t$ is the set off all possible

histories of length $|H| \leq T$ and define $H_{0:k} := (S_0, A_0, \ldots, S_k) \in \mathcal{H}_k$ as the $k$-length subsequence of a given history $H$. We formalize the mutual consistency notion by defining a *risk level assignment* $\mathcal{Y} : \mathcal{H} \rightarrow (0, 1]$ that maps any history $H$ to a risk level $Y$. A risk level assignment $\mathcal{Y}$ is consistent with respect to trajectory perturbations $\xi$ if it enforces the correct total perturbations on all histories, while also respecting all stepwise constraints on the CVaR risk envelope and maintaining the correct sampled action from the risk-dependent policy. We will refer to these sets of constraints as the *risk-assignment consistency constraints*.

**Definition 4** (Risk assignment consistency constraints). *For a given risk-dependent policy $\tilde{\pi}$, initial risk level $\alpha \in (0, 1]$, and history perturbations $\xi \in \Xi_\alpha \left( P^{\tilde{\pi}_h^\alpha} \right)$, a risk level assignment $\mathcal{Y}$ is* consistent *if, for all histories $H \in \mathcal{H}_T$ with $P^{\tilde{\pi}_h^\alpha}(H) > 0$, it satisfies the following constraints:*

1. **Risk propagation:** *The assignment must propagate risk according to $\xi$, that is $\mathcal{Y}(H_{0:0}) = \alpha$ and $\mathcal{Y}(H) = \alpha \cdot \xi(H)$.*

2. **State-level risk envelope:** *For $t \in \{0, \ldots, T-1\}$, the risk envelope constraint over states must be respected for all possible states:*

$$\sum_{s' \in \mathcal{S}} P(s'|S_t, A_t) \frac{\mathcal{Y}(H_{0:t} \cup (A_t, s'))}{\mathcal{Y}(H_{0:t})} = 1.$$

3. **Action-selection consistency:** *For $t \in \{0, \ldots, T-1\}$, the actions taken in the history must match the risk-dependent policy's output for the assigned risk level:*

$$\tilde{\pi}(S_t, \mathcal{Y}(H_{0:t})) = A_t.$$

We are now prepared to formally connect the risk-assignment consistency constraints with the realizability property introduced earlier.

**Lemma 2** (Consistency if and only if realizability). *Under the conditions of Assumptions 1, , for any risk-dependent policy $\tilde{\pi}$ and initial risk level $\alpha \in (0, 1]$, history perturbations $\xi \in \Xi_\alpha \left( P^{\tilde{\pi}_h^\alpha} \right)$ are realizable if and only if there exists a consistent risk level assignment $\mathcal{Y}$ such that all risk-assignment consistency constraints (Def. 4) hold simultaneously.*

The difficulty in satisfying the risk-assignment consistency constraints (Def. 4) lies in finding an assignment $\mathcal{Y}$ that satisfies all three constraint sets *simultaneously*. While it is clear that each constraint set can be satisfied in isolation, their intersection may be empty. This tension is the fundamental source of the CVaR evaluation gap (Eq. 7), which we formalize in the following theorem.

**Theorem 2** (Conditions for CVaR evaluation gap). *Under the conditions of Assumptions 1, for any risk-dependent policy $\tilde{\pi}$ and initial risk level $\alpha \in (0, 1]$, we have $\mathrm{CVaR}_\alpha \left[ Z^{\tilde{\pi}_h^\alpha} \right] = \boldsymbol{v}^{\tilde{\pi}}(s_0, \alpha)$ if and only if there exists at least one set of optimal history perturbations $\xi^\star$ solution to the static CVaR evaluation (Eq. 4) such that the risk-assignment constraints (Def. 4) can be satisfied simultaneously.*

Theorem 2 presents a formal characterization of necessary and sufficient conditions for when a CVaR evaluation gap occurs. It provides the valuable insight that, hidden under the mismatch between a risk-dependent policy's value function and its static CVaR lies an unsolvable constraint satisfaction problem on the risk level evolution. More specifically, risk-dependent policies can induce action-selection consistency constraints that cannot be satisfied simultaneously with the other risk propagation and state-level risk envelope requirements, hampering the evaluation of a policy's true CVaR.

The proposed constraint satisfaction perspective also clarifies why the evaluation is always accurate for Markovian policies (Thm. 3.1 in Hau et al. (2023)). For such policies, the action-selection consistency constraint is non-binding, as the policy does not depend on the risk level. A consistent risk-assignment can therefore always be constructed by recursively applying the CVaR decomposition theorem (Thm. 1), guaranteeing the absence of a CVaR evaluation gap, as detailed in the following corollary.

**Corollary 2** (Existence of corresponding risk-dependent policy). *Under the conditions of Assumptions 1, for any Markovian policy $\pi : \mathcal{H} \rightarrow \mathcal{A}$ and initial risk level $\alpha \in (0, 1]$, there exists a risk-dependent policy $\tilde{\pi}$ such that $\mathrm{CVaR}_\alpha \left[ Z^\pi \right] = \boldsymbol{v}^{\tilde{\pi}}(s_0, \alpha)$.*

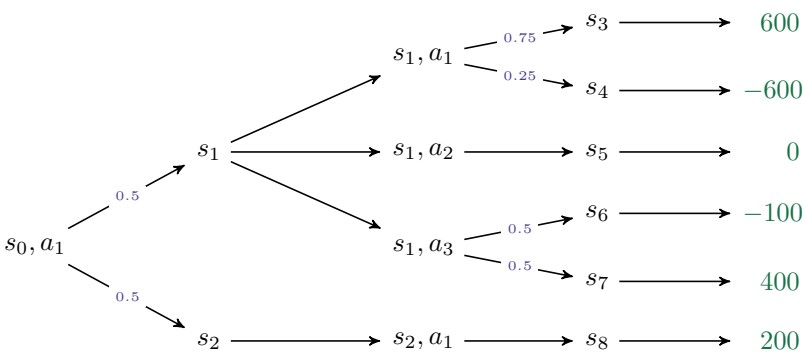

Figure 1: Sample MDP from Hau et al. (2023). Next state transition probabilities are in blue while rewards are in green.

The biggest issue that stems from Theorem 2 and Corollary 2 is that one cannot in general guarantee that *all* risk-dependent policies will have optimal history perturbations $\xi^\star$ that have consistent risk-assignment constraints (Def. 4). As a result, the set of all risk-dependent policies may contain policies with a positive CVaR evaluation gap who will have an inaccurately high $\boldsymbol{v}^{\tilde{\pi}}(s_0, \alpha)$, impeding on the optimality of algorithms searching for $\tilde{\pi}^\star \in \arg\max_{\tilde{\pi}} \boldsymbol{v}^{\tilde{\pi}}(s_0, \alpha)$ like CVaR VI (Chow et al., 2015). We now present a worked example where the optimal risk-dependent policy has a positive CVaR evaluation gap and is therefore suboptimal.

**A deeper dive into the counterexample from Hau et al. (2023)**

We now apply our newly introduced constraint satisfaction analysis to a counterexample presented in Hau et al. (2023). We use the MDP shown in Figure 1, with horizon $T = 2$ and initial risk $\alpha = 0.5$. Note that our MDP differs superficially from the one in Hau et al. (2023), namely because we use a deterministic initial state $s_0$ with a single action available $a_1$ that does not impact the transition to the first state $S_1$, a procedure equivalent to the stochastic initial state presented in the original MDP.

For our example, we consider the risk-dependent policy $\tilde{\pi}$ produced by CVaR VI (Chow et al., 2015) and its optimal state-level perturbations $\tilde{\boldsymbol{\xi}}^\star$ solution to the risk-dependent-policy value function (Eq. 6):

$$\tilde{\pi}(s_1, y) = \begin{cases} a_1 & \text{if } y > 0.5 \\ a_2 & \text{if } y \le 0.5 \end{cases} \qquad \tilde{\boldsymbol{\xi}}^\star(s'|s, y, a) = 1, \quad \text{for all reachable } (s, y, a).$$

The optimal state-level perturbations $\tilde{\boldsymbol{\xi}}^\star$ therefore apply no changes to next states sampled, so the risk level remains $Y_1 = 0.5$ after the transition from $s_0$. As a result, the corresponding history-dependent policy takes action $a_2$ when reaching $s_1$, that is $\tilde{\pi}_h^{0.5}((s_0, a_1, s_1)) = a_2$. Solving the static CVaR evaluation (Eq. 4), we find the corresponding history probabilities and optimal history perturbations $\xi^\star$:

$$P^{\tilde{\pi}_h^{0.5}}(H) = \begin{cases} 0.5 & \text{if } H = (s_0, a_1, s_1, a_2, s_5) \\ 0.5 & \text{if } H = (s_0, a_1, s_2, a_1, s_8) \end{cases} \qquad \xi^\star(H) = \begin{cases} 2 & \text{if } H = (s_0, a_1, s_1, a_2, s_5) \\ 0 & \text{if } H = (s_0, a_1, s_2, a_1, s_8) \end{cases}$$

For the optimal history perturbations $\xi^\star$ to be realizable, there must exist a consistent risk level assignment $\mathcal{Y}$ such that the risk-assignment consistency constraints (Def. 4) have non-empty intersection. Observing that histories of any length are fully defined by their final state since states never repeat in this MDP, the constraints can be expressed as:

1. **Risk propagation**: Directly applying trajectory perturbations $\xi^\star$, we get

$$\mathcal{Y}(s_0) = \alpha = 0.5, \qquad \mathcal{Y}(s_5) = 1, \qquad \text{and} \qquad \mathcal{Y}(s_8) = 0.$$

2. **State-level risk envelope**: For $t = 0$, the constraint is equivalent to $\mathcal{Y}(s_1) + \mathcal{Y}(s_2) = 1$. The constraints for $t = 1$ are $\mathcal{Y}(s_5) = \mathcal{Y}(s_1)$ and $\mathcal{Y}(s_8) = \mathcal{Y}(s_2)$. Combining with the risk propagation

Figure 2: Visual representation of the risk-assignment constraints on the MDP from Figure 1, with the policy $\tilde{\pi}$ obtained using CVaR VI (Chow et al., 2015) and setting $\alpha = 0.5$. Risk propagation constraints are in brown, state-level risk envelope constraints are in purple, and action selection constraints are in teal.

constraint, we find that the risk assignment at $t = 1$ must be

$$\mathcal{Y}(s_1) = 1 \qquad \text{and} \qquad \mathcal{Y}(s_2) = 0. \qquad (10)$$

3. **Action-selection consistency**: The history that ends in $s_5$ requires that at state $s_1$, the action $a_2$ was selected. According to the policy $\tilde{\pi}(s_1, y)$, this is only possible if the risk level satisfies

$$\mathcal{Y}(s_1) \le 0.5. \qquad (11)$$

Figure 2 displays a visual representation of the risk-assignment constraints on the considered MDP given the risk-dependent policy $\tilde{\pi}$ obtained using CVaR VI and an initial risk level $\alpha = 0.5$. We observe that the state-level risk envelope (Eq. 10) and the action-selection consistency (Eq. 11) constraints are impossible to satisfy simultaneously in state $s_1$. Thus, no consistent risk level assignment $\mathcal{Y}$ exists for $\xi^\star$. By Theorem 2, this confirms a positive CVaR evaluation gap at $\alpha = 0.5$, providing context to previous empirical results (Hau et al., 2023).

## 5 From Impossible Evaluation to Impossible Uniform Optimality

Leveraging the risk-assignment constraints perspective developed in the previous section, we now show that there exists an MDP where it is impossible for a single risk-dependent policy $\tilde{\pi}$ to be *uniformly optimal*, that is, being optimal for all initial risk levels $\alpha \in (0, 1]$ simultaneously. This suggests that the limitations of the dual CVaR decomposition are fundamental to the current problem formulation and cannot be solved by simple algorithmic improvements. We begin by formalizing the notion of a uniformly optimal policy.

**Definition 5** (Uniformly optimal policy). *A risk-dependent policy $\tilde{\pi}$ is* uniformly optimal *if its corresponding history-dependent policy $\tilde{\pi}_h^\alpha$ is optimal for all initial risk levels $\alpha \in (0, 1]$. That is, for all $\alpha \in (0, 1]$, we have*

$$\text{CVaR}_\alpha \left[ Z^{\tilde{\pi}_h^\alpha} \right] = \max_{\pi_h} \text{CVaR}_\alpha \left[ Z^{\pi_h} \right].$$

To simplify our argument, we will assume without loss of generality that there is always a single optimal history-dependent policy for every $\alpha$. That is $\arg\min_{\pi_h} \text{CVaR}_\alpha \left[ Z^{\pi_h} \right]$ is always a singleton, where we break ties consistently for policies with the same CVaR values.

For a policy $\tilde{\pi}$ to achieve uniform optimality, its selected actions must align with those of the optimal history-dependent policy $\pi_{h,\alpha}^\star$ for every value of $\alpha$ and all possible histories. By Corollary 2, each optimal policy $\pi_{h,\alpha}^\star$ induces a unique risk-dependent policy $\tilde{\pi}_\alpha$ and risk level assignment $\mathcal{Y}_\alpha$. To be uniformly optimal, a policy $\tilde{\pi}$ therefore has to ensure it simultaneously follows every $\tilde{\pi}_\alpha$ alongside its risk level assignment $\mathcal{Y}_\alpha$. These requirements can be grouped in a set of *optimal-action-selection constraints*.

**Proposition 2** (Uniform optimality constraints). *A risk-dependent policy $\tilde{\pi}$ is uniformly optimal if and only if it simultaneously satisfies all the* optimal-action-selection constraints

$$\tilde{\pi}\left( S_t, \mathcal{Y}_\alpha(H_{0:t}) \right) = \pi_{h,\alpha}^\star(H_{0:t}),$$

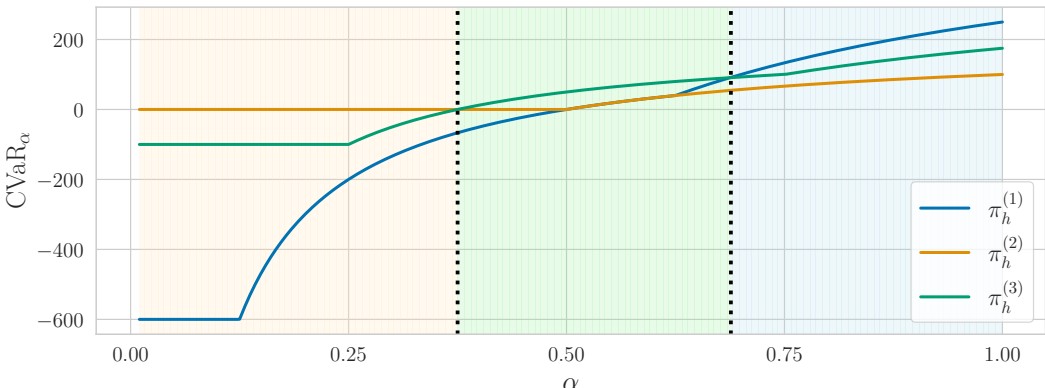

Figure 3: Evolution of the static CVaR evaluation of all policies on the MDP presented in Figure 1 at different initial risk levels $\alpha$. Shaded regions represent the optimal policy $\pi_h^\star$ at a given initial risk level $\alpha$.

defined for all initial risk levels $\alpha \in (0,1]$, optimal policies $\pi_{h,\alpha}^\star$, histories $H \in \mathcal{H}_T$ with $P^{\pi_{h,\alpha}^\star}(H) > 0$, and time steps $t = 0, \ldots, T-1$.

Proposition 2 provides a clear test for uniform optimality: one must check if the set of optimal-action-selection constraints is feasible on an MDP to know whether or not there exists a uniformly optimal policy. If, for two different initial risk levels, the respective optimal policies generate the same risk-augmented state $(S, Y)$ but require different actions, then no single deterministic policy $\tilde{\pi}$ can satisfy all constraints simultaneously. This conflict is the basis for the following impossibility result.

**Theorem 3** (Impossible uniform optimality counterexample). *There exists an MDP satisfying Assumptions 1 for which no single risk-dependent policy $\tilde{\pi} : \mathcal{S} \times (0,1] \to \mathcal{A}$ is uniformly optimal.*

*Proof.* We prove the result by providing an example MDP for which uniform optimality is impossible. We once again use the MDP from Hau et al. (2023), displayed in Figure 1, with horizon $T = 2$. This MDP contains only three different history-dependent policies, which are all fully characterized by the selected action in state $s_1$. We can therefore easily compute the static CVaR of history-dependent policies to see which one is optimal at different initial risk level $\alpha$. To simplify notation, let us denote the three history-dependent policies $\pi_h^{(i)}$ to indicate which action $a_i$ they select in $s_1$. Figure 3 shows the $\mathrm{CVaR}_\alpha$ of each history-dependent policy based on the initial risk level $\alpha$. We can deduce the optimal policy:

$$\pi_{h,\alpha}^\star := \begin{cases} \pi_h^{(2)} & \text{if} \quad \alpha \in [0, 0.375) \\ \pi_h^{(3)} & \text{if} \quad \alpha \in [0.375, 0.6875) \\ \pi_h^{(1)} & \text{if} \quad \alpha \in [0.6875, 1], \end{cases}$$

meaning that the optimal risk-seeking (*alpha* close to 1) choice is to select action $a_1$, while the optimal risk-averse (*alpha* close to 0) choice is instead to pick $a_2$, with $a_3$ being the optimal choice at a moderate risk-level.

Corollary 2 applies to every policy $\pi_h^{(i)}$, hence computing their respective value functions (Eq. 6) grants us state-level perturbations $\tilde{\boldsymbol{\xi}}$ from which we can extract a consistent risk assignment. Combining these risk assignments with the $\alpha$ values where each $\pi_h^{(i)}$ is optimal, we can extract the optimal-action-selection constraints. To streamline our argument, let us consider the state-level risk envelope constraints imposed at state $s_1$ for cases $\alpha = 0.25$, $\alpha = 0.5$, and $\alpha = 0.75$ in particular. For these, the optimal-action-selection-

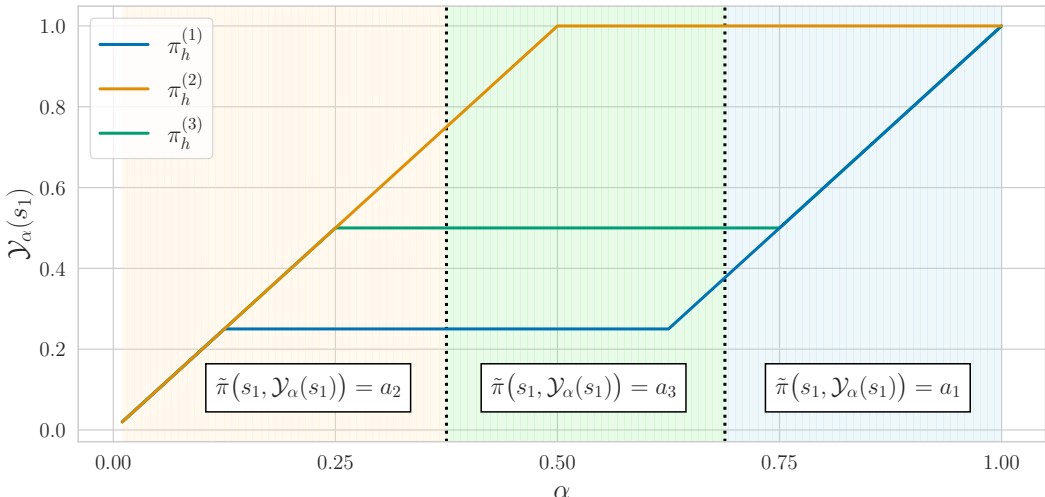

Figure 4: Relation between initial risk level $\alpha$ and the corresponding risk level $\mathcal{Y}_\alpha(s_1)$ for all three possible policies on the MDP presented in Figure 1. Colored areas indicate the optimal policies, with the resulting constraint on $\tilde{\pi}$ displayed explicitly.

constraints in $s_1$ yield:

$$
\begin{aligned}
\pi^\star_{h,0.25} = \pi^{(2)}_h &\implies \mathcal{Y}_{0.25}(s_1) = 0.5 &\implies \tilde{\pi}^\star(s_1, 0.5) = a_2 \\
\pi^\star_{h,0.5} = \pi^{(3)}_h &\implies \mathcal{Y}_{0.5}(s_1) = 0.5 &\implies \tilde{\pi}^\star(s_1, 0.5) = a_3 \\
\pi^\star_{h,0.75} = \pi^{(1)}_h &\implies \mathcal{Y}_{0.75}(s_1) = 0.5 &\implies \tilde{\pi}^\star(s_1, 0.5) = a_1
\end{aligned}
$$

That is, in order to be simultaneously optimal for initial risk levels $\alpha = 0.25$, $\alpha = 0.5$, and $\alpha = 0.75$, a risk-dependent policy $\tilde{\pi}$ is required to take *all* actions $a_1$, $a_2$, and $a_3$ in state $S_t = s_1$ when the risk level is $Y_t = 0.5$, proving the impossibility of uniform optimality. $\qquad\square$

Note that the range of values preventing the existence of a uniformly optimal policy in the MDP presented in Figure 1 extends beyond the $\alpha$ values presented in the proof of Theorem 3. To illustrate this, Figure 4 displays the optimal-action-selection constraints for all $\alpha \in (0, 1]$. Colored regions indicate which policy is considered optimal for the given initial risk level $\alpha$, with a box highlighting the explicit constraint for every region. For every policy, the risk-assignment in $s_1$ ($\mathcal{Y}_\alpha(s_1)$) it corresponds to for an initial risk-level $\alpha$ is shown in a separate color. The figure highlights the presence of a wide range of initial risk levels $\alpha$ where the optimal-action-selection constraints overlap, precluding the presence of a single optimal risk-dependent policy $\pi$ simultaneously optimal for these initial risk levels.

## 6 Conclusion

In this work, we diagnosed the root cause of failures in the static dual CVaR dynamic program decomposition. We started by framing the problem of evaluating a policy's static CVaR as two distinct but related optimization tasks: one over history-level perturbations for the true static CVaR and another over state-level perturbations for the dynamic program. This perspective revealed that a CVaR evaluation gap arises precisely when a set of risk-assignment consistency constraints (Def. 4) have an empty intersection (Thm. 2). Our findings revealed that all history-dependent policies enjoy a null CVaR evaluation gap, which is unfortunately not the case for risk-dependent policies. Building on our risk-assignment constraints, we proved that the dual decomposition itself is fundamentally limited by identifying an MDP where no single risk-dependent policy can be uniformly optimal for all initial risk levels, as the action requirements for optimality at different risk levels become contradictory (Thm. 3).

**Future Directions** Our findings show that seeking a single, uniformly optimal policy with the current dual decomposition approach is flawed. However, for each Markovian policy and initial risk level $\alpha$, there exists a risk-dependent policy with the corresponding CVaR evaluation. This opens a new question: How do we find such *good* risk-dependent policies? Future work should therefore pivot towards developing algorithms that find the optimal policy for a specific initial risk level, similar to methods used for primal-based decompositions (Bäuerle & Ott, 2011). Another alternative research direction is to identify the conditions on the MDP or the policy class under which the risk-assignment consistency constraints are always satisfiable, thereby guaranteeing the absence of an evaluation gap.

Ultimately, despite the identified limitations regarding uniform optimality, the static CVaR dual decomposition retains significant practical appeal. In contrast to primal-based methods, which necessitate augmenting states with unbounded cumulative returns, the dual formulation operates within the bounded $(0, 1]$ risk interval. This compactness allows for a standardized, domain-independent discretization strategy that is more tractable in practice. Ultimately, our analysis provides a foundation for further characterizing, and potentially overcoming, the challenges of risk-averse reinforcement learning. By formally characterizing the evaluation gap, our work identifies the precise constraints that any valid solution must satisfy. This analysis provides the necessary theoretical pathway for future research to derive exact, target-specific dual algorithms that reconcile these computational benefits with rigorous optimality guarantees.

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

# A    Notation

| Symbol | Description |
|---|---|
| $\mathcal{S}$ | Finite state space |
| $\mathcal{A}$ | Finite action space |
| $\Delta(\cdot)$ | Probability simplex |
| $P$ | Transition function $P : \mathcal{S} \times \mathcal{A} \mapsto \Delta(\mathcal{S})$ |
| $\mathcal{R}$ | Reward function $\mathcal{R} : \mathcal{S} \times \mathcal{A} \times \mathcal{S} \mapsto [0, R_{\max}]$ |
| $\gamma$ | Discount factor |
| $s_0$ | Initial state |
| $T$ | Trajectory horizon |
| $H$ | History $(S_0, A_0, S_1, \dots)$ |
| $\mathcal{H}_t$ | Set of histories of length $t$ |
| $H_{0:k}$ | $k$-length subsequence of history $H$ |
| $\pi_h$ | History-dependent policy $\pi_h : \mathcal{H} \mapsto \mathcal{A}$ |
| $\pi$ | Markovian policy $\pi : \mathcal{S} \mapsto \mathcal{A}$ |
| $P^{\pi_h}(H)$ | Probability of history $H$ when following policy $\pi_h$ |
| $\mathcal{R}_{0:T}^H$ | Return of history $H$ |
| $Z^{\pi_h}$ | Random return when sampling histories by following policy $\pi_h$ |
| $\Xi_\alpha(\mathbb{P})$ | CVaR$_\alpha$ risk envelope around probability distribution $\mathbb{P}$ |
| $\xi$ | History-level perturbations $\xi : \mathcal{H} \mapsto [0, \frac{1}{\alpha}]$, $\xi \in \Xi_\alpha(P^\pi)$ |
| $Y$ | Running risk level $Y \in (0, 1]$ |
| $\tilde{\mathcal{S}}$ | Risk-augmented state space $\tilde{\mathcal{S}} := \mathcal{S} \times (0, 1]$ |
| $\tilde{\pi}$ | Risk-dependent policy $\tilde{\pi} : \tilde{\mathcal{S}} \mapsto \mathcal{A}$ |
| $\tilde{\boldsymbol{\xi}}$ | State-level perturbations $\tilde{\boldsymbol{\xi}}(s, a, y) \in \Xi_y(P(\cdot|s, a))$ |
| $\tilde{\boldsymbol{\Xi}}$ | All valid state-level perturbations $\tilde{\boldsymbol{\Xi}} := \big\{ \tilde{\boldsymbol{\xi}} \big| \tilde{\boldsymbol{\xi}}(\cdot|s, y, a) \in \Xi_y\left(P(\cdot|s, a)\right) \forall (s, y, a) \in \mathcal{S} \times (0, 1] \times \mathcal{A} \big\}$ |
| $\tilde{\pi}_h^\alpha$ | History-dependent correspondance of risk-dependent policy $\tilde{\pi}$ at initial risk level $\alpha \in (0, 1]$ |
| $\boldsymbol{v}^{\tilde{\pi}, \tilde{\boldsymbol{\xi}}}$ | Policy-perturbations value function $\boldsymbol{v}^{\tilde{\pi}, \tilde{\boldsymbol{\xi}}} : \mathcal{S} \times (0, 1] \mapsto \mathbb{R}$ |
| $\zeta_\alpha^{\tilde{\boldsymbol{\xi}}}$ | Perturbation mapping $\zeta_\alpha^{\tilde{\boldsymbol{\xi}}} \in \Xi_\alpha(P^{\pi_h})$ |
| $\mathcal{Y}$ | risk level assignment $\mathcal{Y} : \mathcal{H} \mapsto (0, 1]$ |

Table 1: List of notations

# B Omitted proofs

We now restate the results from the main body and present their omitted proofs.

## B.1 Results of Section 3

**Lemma 1** (Value function evaluation). *Under the conditions of Assumptions 1, the value function evaluation of a risk-dependent policy $\tilde{\pi}$ (Eq. 6) is equivalent to solving*

$$\boldsymbol{v}^{\tilde{\pi}}(s, y) = \min_{\tilde{\boldsymbol{\xi}} \in \tilde{\boldsymbol{\Xi}}} \boldsymbol{v}^{\tilde{\pi}, \tilde{\boldsymbol{\xi}}}(s, y), \tag{9}$$

*where the above holds for all state-risk level pairs $(s, y)$ simultaneously, meaning a single state-level perturbations set $\tilde{\boldsymbol{\xi}}^{\star}$ is optimal for all $(s, y)$.*

***Proof outline.*** The result is proven by contradiction. We assume that there exist state-level perturbations yielding a value strictly lower than the minimum defined by the dynamic program at the final time step at any state-risk level pair $(s, y)$. We then demonstrate that due to the recursive nature of the value function (Def. 2), this strict inequality must propagate backwards through time, ultimately implying that the inequality must hold at time $t = 0$, contradicting the uniform initialization of the value function to zero.

*Proof.* We prove the result by contradiction. First observe that by definition of $\boldsymbol{v}^{\tilde{\pi}}$, there must exist $\boldsymbol{\xi} \in \tilde{\boldsymbol{\Xi}}$ such that $\boldsymbol{v}^{\tilde{\pi}} := \boldsymbol{v}^{\tilde{\pi}, \tilde{\boldsymbol{\xi}}}$. Let $\tilde{\boldsymbol{\xi}}^{\star} := \arg\min_{\tilde{\boldsymbol{\xi}} \in \tilde{\boldsymbol{\Xi}}} \boldsymbol{v}^{\tilde{\pi}, \tilde{\boldsymbol{\xi}}}(s, y)$ and suppose there exists $\tilde{\boldsymbol{\xi}}' \in \tilde{\boldsymbol{\Xi}}$ and $(s, y)$ such that $\boldsymbol{v}^{\tilde{\pi}, \tilde{\boldsymbol{\xi}}'}(s, y) < \boldsymbol{v}^{\tilde{\pi}, \tilde{\boldsymbol{\xi}}^{\star}}(s, y)$. Then, by the definition of $\boldsymbol{v}^{\tilde{\pi}, \tilde{\boldsymbol{\xi}}^{\star}}(s, y)$ (Eq. 8), we must have

$$\boldsymbol{v}_T^{\tilde{\pi}, \tilde{\boldsymbol{\xi}}'}(s, y) < \boldsymbol{v}_T^{\tilde{\pi}, \tilde{\boldsymbol{\xi}}^{\star}}(s, y).$$

Let $a = \tilde{\pi}(s, y)$. Expanding the LHS using the definition of the policy-perturbations value function (Eq. 8) and the RHS using the dynamic program definition (Eq. 6), we get

$$\sum_{s'} P(s'|s, a) \tilde{\boldsymbol{\xi}}'(s'|s, y, a) \left[ \mathcal{R}(s, a, s') + \gamma \boldsymbol{v}_{T-1}^{\tilde{\pi}, \tilde{\boldsymbol{\xi}}'}(s', y') \right]$$
$$< \min_{\tilde{\xi} \in \Xi_y(P(\cdot|s, a))} \sum_{s'} P(s'|s, a) \tilde{\xi}(s') \left[ \mathcal{R}(s, a, s') + \gamma \boldsymbol{v}_{T-1}^{\tilde{\pi}, \tilde{\boldsymbol{\xi}}^{\star}}(s', y') \right].$$

Since $\tilde{\boldsymbol{\xi}}'(\cdot|s, y, a) \in \Xi_y(P(\cdot|s, a))$, the above minimum is upper bounded by substituting $\tilde{\boldsymbol{\xi}}'$ directly:

$$\min_{\tilde{\xi} \in \Xi_y(P(\cdot|s, a))} \sum_{s'} P(s'|s, a) \tilde{\xi}(s') \left[ \mathcal{R}(s, a, s') + \gamma \boldsymbol{v}_{T-1}^{\tilde{\pi}, \tilde{\boldsymbol{\xi}}^{\star}}(s', y') \right]$$
$$\leq \sum_{s'} P(s'|s, a) \tilde{\boldsymbol{\xi}}'(s'|s, y, a) \left[ \mathcal{R}(s, a, s') + \gamma \boldsymbol{v}_{T-1}^{\tilde{\pi}, \tilde{\boldsymbol{\xi}}^{\star}}(s', y') \right].$$

Combining these inequalities and canceling the common terms, we obtain

$$\sum_{s'} P(s'|s, a) \tilde{\boldsymbol{\xi}}'(s'|s, y, a) \boldsymbol{v}_{T-1}^{\tilde{\pi}, \tilde{\boldsymbol{\xi}}'}(s', y') < \sum_{s'} P(s'|s, a) \tilde{\boldsymbol{\xi}}'(s'|s, y, a) \boldsymbol{v}_{T-1}^{\tilde{\pi}, \tilde{\boldsymbol{\xi}}^{\star}}(s', y').$$

For the above inequality to hold, there must exist at least one $(s', y')$ such that $\boldsymbol{v}_{T-1}^{\tilde{\pi}, \tilde{\boldsymbol{\xi}}'}(s', y') < \boldsymbol{v}_{T-1}^{\tilde{\pi}, \tilde{\boldsymbol{\xi}}^{\star}}(s', y')$.

Repeating this argument $T$ times implies the existence of $(s, y)$ such that $\boldsymbol{v}_0^{\tilde{\pi}, \tilde{\boldsymbol{\xi}}'}(s, y) < \boldsymbol{v}_0^{\tilde{\pi}, \tilde{\boldsymbol{\xi}}^{\star}}(s, y)$, which is impossible because they all are initialized to 0. $\square$

**Proposition 1** (State-level perturbations evaluation correspondence). *Under the conditions of Assumptions 1, for any risk-dependent policy $\tilde{\pi}$, initial risk level $\alpha$, and state-level perturbations $\tilde{\boldsymbol{\xi}} \in \tilde{\boldsymbol{\Xi}}$, the mapping $\zeta_\alpha^{\tilde{\boldsymbol{\xi}}}$ produces valid history perturbations, that is $\zeta_\alpha^{\tilde{\boldsymbol{\xi}}} \in \Xi_\alpha\left(P^{\tilde{\pi}_h^\alpha}\right)$, for which we have*

$$\sum_{H \in \mathcal{H}_T} P^{\tilde{\pi}_h^\alpha}(H)\,\zeta_\alpha^{\tilde{\boldsymbol{\xi}}}(H)\,\mathcal{R}_{0:T}^H = \boldsymbol{v}^{\tilde{\pi},\tilde{\boldsymbol{\xi}}}(s_0, \alpha).$$

***Proof outline.*** The proof proceeds in two steps. First, we establish that the mapping $\zeta_\alpha^{\tilde{\boldsymbol{\xi}}}$ produces valid history perturbations (i.e., within the risk envelope) by verifying that the cumulative product of state perturbations satisfies the required probability and boundedness constraints. Second, we prove the equality between the history-level summation and the recursive value function. This is achieved via backward induction, showing that the single-step updates aggregate exactly to the return weighted by the constructed history perturbations.

*Proof.* We first prove $\zeta_\alpha^{\tilde{\boldsymbol{\xi}}} \in \Xi_\alpha\left(P^{\tilde{\pi}_h^\alpha}\right)$ and then we prove $\sum_{H \in \mathcal{H}_T} P^{\tilde{\pi}_h^\alpha}(H)\,\zeta_\alpha^{\tilde{\boldsymbol{\xi}}}(H)\,\mathcal{R}_{0:T}^H = \boldsymbol{v}^{\tilde{\pi},\tilde{\boldsymbol{\xi}}}(s_0, \alpha)$.

**Proof of $\zeta_\alpha^{\tilde{\boldsymbol{\xi}}} \in \Xi_\alpha\left(P^{\tilde{\pi}_h^\alpha}\right)$.** First observe that, when fixing $\tilde{\boldsymbol{\xi}}$ and $\alpha$, the resulting risk level at time $t$ for history $H \in \mathcal{H}_T$ is always given recursively by $Y_t(H|\tilde{\boldsymbol{\xi}}, \alpha) = \alpha \prod_{\tau=1}^{t-1} \tilde{\boldsymbol{\xi}}(S_{\tau+1}|S_\tau, A_\tau, Y_\tau)$ with initial condition $Y_0 = \alpha$. To alleviate notation, we will mute these dependencies and simply write $Y_t$ when $H$, $\tilde{\boldsymbol{\xi}}$, and $\alpha$ are clear from context. Also note that we can exclude trajectories $H$ such that $P^{\tilde{\pi}_h^\alpha}(H) = 0$ from our analysis because these do not induce any constraint in the definition of $\Xi_\alpha\left(P^{\tilde{\pi}_h^\alpha}\right)$ and they are always excluded from the computation of $\boldsymbol{v}^{\tilde{\pi},\tilde{\boldsymbol{\xi}}}$. As a result, in the rest of the proof we can always assume that we have $A_t = \tilde{\pi}(S_t, Y_T)$ when iterating over a history's actions.

For a fixed $H \in \mathcal{H}_T$ and $\alpha$, we have by definition $\zeta_\alpha^{\tilde{\boldsymbol{\xi}}}(H) = \prod_{t=0}^{T-1} \tilde{\boldsymbol{\xi}}(S_{t+1}|S_t, Y_t, A_t)$, which is nothing more than $Y_T/\alpha$ from our above remark. Because $\tilde{\boldsymbol{\xi}}(S_t|S_{t-1}, Y_{t-1}, A_{t-1}) \in [0, 1/Y_{t-1}]$ for all $t = 1, \ldots, T$, we also have that $Y_t = Y_{t-1}\tilde{\boldsymbol{\xi}}(S_t|S_{t-1}, Y_{t-1}, A_{t-1}) \in (0, 1]$. Hence we have in particular $Y_T \in (0, 1]$. From the observation that $Y_T = \alpha\zeta_\alpha^{\tilde{\boldsymbol{\xi}}}(H)$, it follows that $\zeta_\alpha^{\tilde{\boldsymbol{\xi}}}(H) \in [0, 1/\alpha]$ (i).

To prove $\sum_{H \in \mathcal{H}_T} P^{\tilde{\pi}_h^\alpha}(H)\,\zeta_\alpha^{\tilde{\boldsymbol{\xi}}}(H) = 1$, we proceed by backward induction:

$$\sum_{H \in \mathcal{H}_T} P^{\tilde{\pi}_h^\alpha}(H)\,\zeta_\alpha^{\tilde{\boldsymbol{\xi}}}(H) = \sum_{H \in \mathcal{H}_T} \prod_{t=0}^{T-1} P(S_{t+1}|S_t, A_t)\,\tilde{\boldsymbol{\xi}}(S_{t+1}|S_t, Y_t, A_t)$$

$$\overset{(a)}{=} \sum_{H \in \mathcal{H}_{T-1}} \sum_{S_T \in \mathcal{S}} \prod_{t=0}^{T-1} P(S_{t+1}|S_t, A_t)\,\tilde{\boldsymbol{\xi}}(S_{t+1}|S_t, Y_t, A_t)$$

$$\overset{(b)}{=} \sum_{H \in \mathcal{H}_{T-1}} \prod_{t=0}^{T-2} P(S_{t+1}|S_t, A_t)\,\tilde{\boldsymbol{\xi}}(S_{t+1}|S_t, Y_t, A_t) \sum_{S_T \in \mathcal{S}} P(S_T|S_{T-1}, A_{T-1})\,\tilde{\boldsymbol{\xi}}(S_T|S_{T-1}, Y_{T-1}, A_{T-1})$$

$$\overset{(c)}{=} \sum_{H \in \mathcal{H}_{T-1}} \prod_{t=0}^{T-2} P(S_{t+1}|S_t, A_t)\,\tilde{\boldsymbol{\xi}}(S_{t+1}|S_t, Y_t, A_t)$$

For step (a) we used the observation that the set of histories of length $T$ is the set of histories at time $T-1$, upon which we concatenate the action $A_{T-1}$ selected by $\tilde{\pi}_h^\alpha$ and the possible next states $S_T$. In step (b) we exploited the fact that we can peel off the terms depending on $S_T$ from the product. Step (c) exploited the fact that $\tilde{\boldsymbol{\xi}}(\cdot|s, y, a) \in \Xi_y\left(P(\cdot|s, a)\right)$ for all $a = \tilde{\pi}(s, y)$. Repeating the above manipulation $T$ times yields

$$\sum_{H \in \mathcal{H}_T} P^{\tilde{\pi}_h^\alpha}(H)\,\zeta_\alpha^{\tilde{\boldsymbol{\xi}}}(H) = \cdots = \sum_{S_1 \in \mathcal{S}} P(S_1|S_0, A_0)\,\tilde{\boldsymbol{\xi}}(S_1|S_0, Y_0, A_0) = 1,$$

, where we used the fact that $\tilde{\boldsymbol{\xi}}(\cdot|S_0, Y_0, A_0) \in \Xi_{Y_0}\left(P(\cdot|S_0, A_0)\right)$ for the last equality, yielding the desired property (ii). Combining properties (i) and (ii) proves $\zeta_\alpha^{\tilde{\boldsymbol{\xi}}} \in \Xi_\alpha\left(P^{\tilde{\pi}_h^\alpha}\right)$.

**Proof of** $\sum_{H\in\mathcal{H}_T} P^{\tilde{\pi}_h^\alpha}(H)\,\zeta_\alpha^{\tilde{\boldsymbol{\xi}}}(H)\,\mathcal{R}_{0:T}^H = \boldsymbol{v}^{\tilde{\pi},\tilde{\boldsymbol{\xi}}}(s_0,\alpha).$ We now prove the stated equality by proving a slightly more general result. Let $\mathcal{H}_T^s$ denote the set of all histories $H\in\mathcal{H}_T$ beginning with $S_0 = s$. We shall prove that the following result holds for all $s\in\mathcal{S}$ and $\alpha\in(0,1]$:

$$\sum_{H\in\mathcal{H}_T^s} P^{\tilde{\pi}_h^\alpha}(H)\,\zeta_\alpha^{\tilde{\boldsymbol{\xi}}}(H)\,\mathcal{R}_{0:T}^H = \boldsymbol{v}^{\tilde{\pi},\tilde{\boldsymbol{\xi}}}(s,\alpha).$$

We proceed by induction.

**Base case** $t = 1$. Let $\alpha\in(0,1]$, $s\in\mathcal{S}$ and set $Y_0 = \alpha$, we get

$$\begin{aligned}
\boldsymbol{v}_1^{\tilde{\pi},\tilde{\boldsymbol{\xi}}}(s,Y_0) &= \sum_{s'\in\mathcal{S}} P(s'|s,A_0)\,\tilde{\boldsymbol{\xi}}(s'|s,Y_0,A_0)\left[\mathcal{R}(s,A_0,s') + \gamma\boldsymbol{v}_0^{\tilde{\pi}}(s',y')\right] \\
&= \sum_{s'\in\mathcal{S}} P(s'|s,A_0)\,\tilde{\boldsymbol{\xi}}(s'|s,Y_0,A_0)\,\mathcal{R}(s,A_0,s') \\
&= \sum_{H\in\mathcal{H}_1^s} P^{\tilde{\pi}_h^\alpha}(H)\,\zeta_\alpha^{\tilde{\boldsymbol{\xi}}}(H)\,\mathcal{R}_{0:1}^H,
\end{aligned}$$

where we used $A_0 = \tilde{\pi}(s,Y_0)$ and $y' = Y_0\,\tilde{\boldsymbol{\xi}}(s'|s,Y_0,A_0)$ to alleviate notation. The result holds for all $s\in\mathcal{S}$ and $\alpha\in(0,1]$.

**Induction step.** Suppose $\boldsymbol{v}_{t-1}^{\tilde{\pi},\tilde{\boldsymbol{\xi}}}(s,\alpha) = \sum_{H\in\mathcal{H}_{t-1}^s} P^{\tilde{\pi}_h^\alpha}(H)\,\zeta_\alpha^{\tilde{\boldsymbol{\xi}}}(H)\,\mathcal{R}_{0:t-1}^H$. Again setting $Y_0 = \alpha$, we get

$$\begin{aligned}
\boldsymbol{v}_t^{\tilde{\pi},\tilde{\boldsymbol{\xi}}}(s,Y_0) &= \sum_{s'\in\mathcal{S}} P(s'|s,A_0)\,\tilde{\boldsymbol{\xi}}(s'|s,Y_0,A_0)\left[\mathcal{R}(s,A_0,s') + \gamma\boldsymbol{v}_{t-1}^{\tilde{\pi}}(s',y')\right] \\
&= \sum_{s'\in\mathcal{S}} P(s'|s,A_0)\,\tilde{\boldsymbol{\xi}}(s'|s,Y_0,A_0)\,\mathcal{R}(s,A_0,s') + \gamma\sum_{H'\in\mathcal{H}_{t-1}^{s'}} P^{\tilde{\pi}_h^{y'}}(H')\,\zeta_{y'}(\tilde{\boldsymbol{\xi}})(H')\,\mathcal{R}_{0:t-1}^{H'} \\
&\stackrel{(a)}{=} \sum_{s'\in\mathcal{S}} P(s'|s,A_0)\,\tilde{\boldsymbol{\xi}}(s'|s,Y_0,A_0)\left[\sum_{H'\in\mathcal{H}_{t-1}^{s'}} P^{\tilde{\pi}_h^{y'}}(H')\,\zeta_{y'}(\tilde{\boldsymbol{\xi}})(H')\left(\mathcal{R}(s,A_0,s') + \gamma\mathcal{R}_{0:t-1}^{H'}\right)\right] \\
&\stackrel{(b)}{=} \sum_{s'\in\mathcal{S}}\sum_{H'\in\mathcal{H}_{t-1}^{s'}} P(s'|s,A_0)\,\tilde{\boldsymbol{\xi}}(s'|s,Y_0,A_0)\,P^{\tilde{\pi}_h^{y'}}(H')\,\zeta_{y'}(\tilde{\boldsymbol{\xi}})(H')\left(\mathcal{R}(s,A_0,s') + \gamma\mathcal{R}_{0:t-1}^{H'}\right) \\
&\stackrel{(c)}{=} \sum_{H\in\mathcal{H}_t^s} P^{\tilde{\pi}_h^\alpha}(H)\,\zeta_\alpha^{\tilde{\boldsymbol{\xi}}}(H)\,\mathcal{R}_{0:t}^H,
\end{aligned}$$

where we used $A_0 = \tilde{\pi}(s,Y_0)$ and $y' = y_0\,\tilde{\boldsymbol{\xi}}(s'|s,Y_0,A_0)$ to alleviate notation. In step (a), we exploited the fact that $\sum_{H'\in\mathcal{H}_{t-1}^{s'}} P^{\tilde{\pi}_h^{y'}}(H')\,\zeta_{y'}(\tilde{\boldsymbol{\xi}})(H') = 1$ to put the $\mathcal{R}(s,A_0,s')$ term inside the sum. Step (b) used the fact that the $P(s'|s,A_0)$ and $\tilde{\boldsymbol{\xi}}(s'|s,Y_0,A_0)$ terms do not rely on $H'$ to move the summation. Finally, step (c) relied on the fact that all trajectories starting from $S_0 = s$ can be expressed as the union over $s'$ of all trajectories trajectories with $S_1 = s'$ and updating the necessary probability, perturbations, and reward terms accordingly. The result follows by setting $S = s_0$ and picking the desired $\alpha\in(0,1]$. $\qquad\square$

## B.2 Results of Section 4

**Lemma 2** (Consistency if and only if realizability). *Under the conditions of Assumptions 1, , for any risk-dependent policy $\tilde{\pi}$ and initial risk level $\alpha\in(0,1]$, history perturbations $\xi\in\Xi_\alpha\left(P^{\tilde{\pi}_h^\alpha}\right)$ are realizable if and only if there exists a consistent risk level assignment $\mathcal{Y}$ such that all risk-assignment consistency constraints (Def. 4) hold simultaneously.*

***Proof outline.*** The proof is constructive for both directions. For the forward direction ( $\implies$ ), given realizable perturbations, we construct the risk assignment $\mathcal{Y}$ recursively from the existing state-level perturbations and verify that this construction inherently satisfies the consistency constraints. For the backward direction ( $\impliedby$ ), we construct the state-level perturbations by taking the ratio of the consistent risk assignments at subsequent steps, showing that the resulting perturbations are valid members of the risk envelope and aggregate to the target history perturbation.

*Proof.* ( $\implies$ ) Assume $\xi$ is realizable by some $\tilde{\boldsymbol{\xi}}$. Define the assignment $\mathcal{Y}(H_t)$ by recursively computing the risk levels generated when following $\tilde{\boldsymbol{\xi}}$, that is $\mathcal{Y}(H_{0:t+1}) = \mathcal{Y}(H_{0:t}) \cdot \tilde{\boldsymbol{\xi}}(S_{t+1}|S_t, \mathcal{Y}(H_{0:t}), A_t)$ for all $t = 0, \ldots, T-1$ and initialized at $\mathcal{Y}(H_{0:0}) = \alpha$. Because $\xi$ is realizable, it follows that $\mathcal{Y}(H) = \zeta_\alpha^{\tilde{\boldsymbol{\xi}}}(H) = \xi(H)$ for all histories $H \in \mathcal{H}_T$, establishing the risk propagation constraint. Now observe that by construction we have:

$$\sum_{s' \in \mathcal{S}} P(s'|S_t, A_t) \frac{\mathcal{Y}(H_{0:t} \cup (A_t, s'))}{\mathcal{Y}(H_{0:t})} = \sum_{s' \in \mathcal{S}} P(s'|S_t, A_t) \, \tilde{\boldsymbol{\xi}}(s'|S_t, \mathcal{Y}(H_{0:t}), A_t).$$

By this observation and the definition of $\tilde{\boldsymbol{\xi}} \in \tilde{\boldsymbol{\Xi}}$, the state-level risk envelope and action selection constraints follow, proving the desired statement.

( $\impliedby$ ) Assume a consistent risk level assignment $\mathcal{Y}$ exists. We construct $\tilde{\boldsymbol{\xi}}$ as follows: for any $(s, y, a)$ reached by a history $H$ with $P^{\tilde{\pi}_h^\alpha}(H) > 0$, that is $s = S_t, y = \mathcal{Y}(H_{0:t}), a = A_t$, define $\tilde{\boldsymbol{\xi}}(s'|s, y, a) \coloneqq \mathcal{Y}(H_t \cup (a, s'))/y$. For all other $(s, y, a)$, define $\tilde{\boldsymbol{\xi}}(s'|s, y, a) = 1$ to trivially ensure it is part of $\Xi_y \left( P(\cdot|s, a) \right)$. By the state-level risk envelope and action selection constraints, it follows that $\tilde{\boldsymbol{\xi}} \in \tilde{\boldsymbol{\Xi}}$. Moreover, for all histories we have $\zeta_\alpha^{\tilde{\boldsymbol{\xi}}}(H) = \prod_{t=0}^{T-1} \frac{\mathcal{Y}(H_{0:t+1})}{\mathcal{Y}(H_{0:t})} = \frac{\mathcal{Y}(H_{0:T})}{\mathcal{Y}(H_{0:0})} = \frac{\alpha \cdot \xi(H)}{\alpha} = \xi(H)$. Thus, $\xi$ is realizable. $\qquad\square$

**Theorem 2** (Conditions for CVaR evaluation gap). *Under the conditions of Assumptions 1, for any risk-dependent policy $\tilde{\pi}$ and initial risk level $\alpha \in (0, 1]$, we have $\mathrm{CVaR}_\alpha \left[ Z^{\tilde{\pi}_h^\alpha} \right] = \boldsymbol{v}^{\tilde{\pi}}(s_0, \alpha)$ if and only if there exists at least one set of optimal history perturbations $\xi^\star$ solution to the static CVaR evaluation (Eq. 4) such that the risk-assignment constraints (Def. 4) can be satisfied simultaneously.*

***Proof outline.*** This proof relies on the equivalence established in Lemma 2 and the upper bound property from Corollary 1. For the forward direction, if the evaluation gap is zero, the optimal history perturbations must be realizable by some state-level perturbations, implying consistency. For the reverse direction, if consistency holds, the optimal history perturbations are realizable. This allows us to exactly match the dynamic program's value between the true static CVaR and the realizable value, proving equality.

*Proof.* ( $\implies$ ) Assume $\mathrm{CVaR}_\alpha \left[ Z^{\tilde{\pi}_h^\alpha} \right] = \boldsymbol{v}^{\tilde{\pi}}(s_0, \alpha)$. Let $\tilde{\boldsymbol{\xi}}^\star$ be a minimizer for $\boldsymbol{v}^{\tilde{\pi}, \tilde{\boldsymbol{\xi}}}(s_0, \alpha)$. From Proposition 1 we know we can compute $\xi' \coloneqq \zeta_\alpha(\tilde{\boldsymbol{\xi}}^\star)$, where $\xi' \in \Xi_\alpha \left( P^{\tilde{\pi}_h^\alpha} \right)$ are valid trajectory perturbations and $\sum_{H \in \mathcal{H}_T} P^{\tilde{\pi}_h^\alpha}(H) \xi'(H) \mathcal{R}_{0:T}^H = \boldsymbol{v}^{\tilde{\pi}, \tilde{\boldsymbol{\xi}}^\star}(s_0, \alpha) = \boldsymbol{v}^{\tilde{\pi}}(s_0, \alpha)$. Since we assumed $\mathrm{CVaR}_\alpha \left[ Z^{\tilde{\pi}_h^\alpha} \right] = \boldsymbol{v}^{\tilde{\pi}}(s_0, \alpha)$, this means $\xi'$ must be an optimal trajectory perturbations set. Because $\xi'$ is realizable by construction, applying Lemma 2 gives the desired result.

( $\impliedby$ ) Let $\xi^\star$ be an optimal trajectory perturbations set for which there exists a consistent risk assignment. By Lemma 2, we know $\xi^\star$ is realizable and hence there exists a $\tilde{\boldsymbol{\xi}}' \in \tilde{\boldsymbol{\Xi}}$ such that $\zeta_\alpha(\tilde{\boldsymbol{\xi}}') = \xi^\star$. From Proposition 1 and the definition of $\xi^\star$, we have $\boldsymbol{v}^{\tilde{\pi}, \tilde{\boldsymbol{\xi}}'}(s_0, \alpha) = \sum_{H \in \mathcal{H}_T} P^{\tilde{\pi}_h^\alpha}(H) \xi^\star(H) \mathcal{R}_{0:T}^H = \mathrm{CVaR}_\alpha \left[ Z^{\tilde{\pi}_h^\alpha} \right]$. Since $\boldsymbol{v}^{\tilde{\pi}}(s_0, \alpha) = \min_{\tilde{\boldsymbol{\xi}}} \boldsymbol{v}^{\tilde{\pi}, \tilde{\boldsymbol{\xi}}}(s_0, \alpha)$, it must be that $\boldsymbol{v}^{\tilde{\pi}}(s_0, \alpha) \leq \boldsymbol{v}^{\tilde{\pi}, \tilde{\boldsymbol{\xi}}^\star}(s_0, \alpha) = \mathrm{CVaR}_\alpha \left[ Z^{\tilde{\pi}_h^\alpha} \right]$. Combining this with the opposite inequality from Corollary 1 yields equality. $\qquad\square$

**Corollary 2** (Existence of corresponding risk-dependent policy). *Under the conditions of Assumptions 1, for any Markovian policy $\pi : \mathcal{H} \to \mathcal{A}$ and initial risk level $\alpha \in (0, 1]$, there exists a risk-dependent policy $\tilde{\pi}$ such that $\mathrm{CVaR}_\alpha \left[ Z^\pi \right] = \boldsymbol{v}^{\tilde{\pi}}(s_0, \alpha)$.*

***Proof outline.*** The result is proven by construction. We show that a Markovian policy can be trivially viewed as a risk-dependent policy where the action choice is independent of the risk level $y$. This renders

the action-selection consistency constraints non-binding. Consequently, the standard CVaR decomposition theorem guarantees the existence of a valid risk assignment, ensuring no gap exists.

*Proof.* We prove the statement by explicitly proving that $\tilde{\pi} = \pi$ is the desired policy. First observe that one can obtain a risk assignment $\mathcal{Y}$ by leveraging the CVaR Decomposition Theorem (Thm. 1 $T$ times, for instance by applying the dynamic programming operator in Equation 1. By construction, this risk assignment will satisfy the risk propagation and state-level risk envelope constraints. Considering $\tilde{\pi} = \pi$, the action-selection consistency constraints are also trivially satisfied, ensuring $\mathcal{Y}$ is a consistent risk assignment mapping from $\tilde{\pi}$ and $\pi$, hence $\tilde{\pi}$ does not have a CVaR evaluation gap as per Theorem 2.

$\square$

### B.3 Results of Section 5

**Proposition 2** (Uniform optimality constraints)**.** *A risk-dependent policy $\tilde{\pi}$ is uniformly optimal if and only if it simultaneously satisfies all the* optimal-action-selection constraints

$$\tilde{\pi}\big(S_t, \mathcal{Y}_\alpha(H_{0:t})\big) = \pi_{h,\alpha}^\star(H_{0:t}),$$

*defined for all initial risk levels $\alpha \in (0,1]$, optimal policies $\pi_{h,\alpha}^\star$, histories $H \in \mathcal{H}_T$ with $P^{\pi_{h,\alpha}^\star}(H) > 0$, and time steps $t = 0, \ldots, T-1$.*

***Proof outline.*** We prove the equivalence by linking the definitions of uniform optimality and action consistency. In the forward direction, uniform optimality implies the policy must match the optimal history-dependent policy for every $\alpha$, which in turn implies matching the actions prescribed by those optimal policies at the reachable risk-augmented states. In the reverse direction, satisfying the constraints ensures the policy mimics the optimal history-dependent policy for every $\alpha$, thereby guaranteeing the CVaR values match.

*Proof.* ( $\implies$ ) Assume $\tilde{\pi}$ is uniformly optimal. By definition, for any $\alpha \in (0,1]$, we have $\mathrm{CVaR}_\alpha\big[Z^{\tilde{\pi}_h^\alpha}\big] = \max_{\pi_h} \mathrm{CVaR}_\alpha\big[Z^{\pi_h}\big] = \mathrm{CVaR}_\alpha\Big[Z^{\pi_{h,\alpha}^\star}\Big]$. Because we assumed there is only one optimal $\pi_h^\star(\alpha)$, it follows that the history distributions $P^{\tilde{\pi}_h^\alpha}$ and $P^{\pi_h^\star(\alpha)}$ must match. Because the history distributions only depend on policies in whether the action $A_t$ is selected, it follows that $\pi_h^\alpha$ and $\pi_h^\star(\alpha)$ always select the same action $A_t$ given $H_{0:t}$. Because of the feasibility of the dynamic program decomposition for $\pi_h(\alpha)$ (Corollary 2), this is equivalent to saying $\tilde{\pi}\big(S_t, \mathcal{Y}_\alpha(H_{0:t})\big) = \pi_{h,\alpha}^\star(H_{0:t})$. As this must hold for all $\alpha \in (0,1]$, $\tilde{\pi}$ must satisfy the optimal-action-selection constraints.

( $\impliedby$ ) Assume $\tilde{\pi}$ satisfies the optimal-action-selection constraints. This means that for any given $\alpha \in (0,1]$ and any history $H_{0:t}$ reachable under the optimal policy $\pi_{h,\alpha}^\star$, the action chosen by $\tilde{\pi}_h^\alpha$ is the same as the action chosen by $\pi_{h,\alpha}^\star$. This ensures that the two policies are identical, $\tilde{\pi}_h^\alpha = \pi_{h,\alpha}^\star$. Consequently, their induced trajectory distributions are identical, $P^{\tilde{\pi}_h^\alpha} = P^{\pi_{h,\alpha}^\star}$. It follows directly that their CVaR evaluations must also be identical:

$$\mathrm{CVaR}_\alpha\Big[Z^{\tilde{\pi}_h^\alpha}\Big] = \mathrm{CVaR}_\alpha\Big[Z^{\pi_{h,\alpha}^\star}\Big] = \max_{\pi_h} \mathrm{CVaR}_\alpha\big[Z^{\pi_h}\big].$$

Since this holds for all $\alpha \in (0,1]$, the policy $\tilde{\pi}$ is uniformly optimal. $\square$

