# OpenReview forum: "On the Fundamental Limitations of Dual Static CVaR Decompositions in Markov Decision Processes"
_TMLR — Rejected by TMLR_

### Review · Reviewer_KU88 · 2025-10-20

**Summary Of Contributions:**

Summary of contributions:
1. This work analyses the reason why static dual CVaR dynamic program decomposition fails.
2. They formulate a history-level pertubation and state-level pertubation optimization formualation for the DP.
3. They highlight the gap in these two formulations and posit that this is due to empty intersection of the risk-assignment constraints pertaining to the formulations.
4. They further highlight the limitation of the dual formulation by giving an example of an MDP where there can be no action-consistent optimal policy for different risk levels.

**Audience:**

Yes

**Audience Explanation:**

The area of Static Risk-Averse Reinforcement Learning is definitely an interesting topic which would be interesting to an audience of RL, risk aware machine learning and safe machine learning researchers. Further it would be useful for future research on dynamic risk-averse RL.

**Claims And Evidence:**

No

**Claims Explanation:**

I have some questions about the theoretical framework, once those are addressed I am open to changing my response.

### On Proofs:

Proof of Lemma 1:
1. How is the min defined in equation 9? What is the nature of the space $\tilde{\Xi}$ or the structure?
2. Since min is monotonous with P and R remaining constant ->. ? monotonous with respect to what?

Proof of Propostion 1:
1. Because $\tilde{\xi}_t \in [0,1/Y_t]$ we also have that $Y_t \in [0,1]$ hence $Y_T (=\Pi \tilde{\xi}_t) \in [0, 1]$ -> If $Y_t$ is in [0,1] then by your statement $\tilde{\xi}_t$ can be greater than 1 and there is no reason why $Y_T <1$ unless I misread something. (Here ${\tilde{\xi}}_t = \tilde{\xi}(S_{t+1}|S_t, Y_t, A_t)$)

2. In the lines that follow some intution (repeated in the proof again for the property ii) would be helpful, what does $\tilde{\xi}$ denote and why does the property hold for a single transition?

3. The proof could use some outline.



Proof of Lemma 2:

1. What does the dagger denote? Please add it to the notation list. Is it a typo and should be $\mathcal{Y}$?

Proof of Corollary 2:

1. what is $\bar{p}$ ?

I would highly recommend adding a proof outline which explains the key mathematical idea used in the proofs and intuitively explains the idea of the proof step by step. This improves the accessibility of the paper.


### On writing and conceptual clarity:

1. While both perturbations impact the sampling of events, $\tilde{\xi}$ also dictates the risk level evolution - please explain and expand more on this.

2. In general I would recommend adding remarks from exisiting text to highlight key insights. Also if perhaps the figure 3 can have either a counterintuitive insight or something meaningful (or perhaps a simple insight even) after the equation.

I am open to having a discussion and during the discussion I might have a few more clarification question.

**Requested Changes:**

I will refine this list of changes once the discussion on the concerns with the claims are addressed. Overall the paper is well-written  and these changes will not be different from the points I raise above.

[Critical] Please address the points I raised above.

[Good to have] It should be made clear that the policies considered in the paper are deterministic.

[Good to have] Please make it clear that the paper considers finite-horizon discounted reward RL

[Nit] For some reason your eqref, citep and ref  (Equations and references) are not hyperlinked. Would be good to have since it makes it easier to navigate.

---

### Review · Reviewer_kvKE · 2025-10-20

**Summary Of Contributions:**

This paper investigates the limitations of dynamic programming approaches based on the dual decomposition of CVaR in MDPs. While prior work had shown counterexamples where dual CVaR dynamic programming fails, the underlying reason remained unclear. The authors make the following contributions:

1. They formalize static CVaR evaluation as two optimization problems (over history-level perturbations and state-level perturbations).
2. They identify a set of constraints that must hold for the two formulations to coincide, and show that inconsistencies explain observed evaluation errors.
3. They define and analyze the discrepancy between dynamic programming evaluation and true static CVaR, proving it corresponds to unsatisfiable consistency constraints.
4. They provide an MDP where no single risk-dependent policy is optimal for all risk levels, proving a fundamental limitation of the dual CVaR decomposition approach.

**Audience:**

Yes

**Audience Explanation:**

I believe this paper will be of interest for advancing both the theoretical foundations and the practical development of risk-aware reinforcement learning.

**Claims And Evidence:**

Yes

**Claims Explanation:**

Overall, the paper makes a strong theoretical contribution by diagnosing the root cause of previously observed failures and by formalizing structural limitations of dual static CVaR decompositions in MDPs

**Requested Changes:**

1. Range of $\alpha$: the authors write $\alpha \in [0, 1]$, i believe it should be $(0, 1]$.
2. Assumptions recap: I believe all the states results are valid under finite state and action space and discrete distribution. Stating all the assumptions explicitly after the main theorems would help the reader.
3. Practical relevance and impact: A discussion of practical consequences of these findings would greatly increase the impact of the work.

---

### Review · Reviewer_7zrp · 2025-12-29

**Summary Of Contributions:**

- The papers provides clearer explanation why static dual CVaR DP decomposition can fail.
- The coined CVaR evaluation gap expresses that the value function computed by dual CVaR DP can systematically overestimate the true static CVaR of the induced policy.
- The authors define risk-assignment consistency constraints which need to be satisfied and connect this finding to prior identified counter cases.
- The paper presents an MDP where no single risk-dependent policy can be optimal for all possible risk levels.

**Audience:**

Yes

**Audience Explanation:**

The papers' findings are relevant for the TMLR audience in that they provide additional explanations why current work on static dual CVaR DP decomposition might provides sub-optimal policies. This strongly supports future work on finding risk-dependent policies for MDPs, i.e., shapes research in safety RL.

**Broader Impact Concerns:**

-

**Claims And Evidence:**

Yes

**Claims Explanation:**

The paper provides clear argumentations and novel explanations why static dual CVaR DP decomposition can lead to sub-optimal risk-dependent policies. The individual components, e.g., CVaR evaluation gap and consistency constraints are thoroughly explained and proven. The paper adds further evalutions by connecting the consistency constraints to already disclosed failure cases of prior work and by presenting an MDP that demonstrates the impossibility of uniform optimality across risk levels.

I only wonder if additional experiments on more MDP counter examples from Hau et al. 2023 could be useful to underline that the chosen MDP is not just an edge case.

**Requested Changes:**

- It would be interesting to hear the authors' perspective if analyzing further counterexample MDPs from Hau et al. 2023 could be useful to show that only isolated cases might be problematic.

---

> ### Author Response · Authors · 2026-01-23
> **Response to Reviewer**
>
> We thank the reviewer for their comments.
>
> > It would be interesting to hear the authors' perspective if analyzing further counterexample MDPs from Hau et al. 2023 could be useful to show that only isolated cases might be problematic.
>
> Similar issues arise in the variant couterexample MDPs from Hau et al., 2023. For instance, Figure 1 in our article is equivalent to setting M=600 in the MDP in Figure 4 of Hau et al., 2023. Setting other values of M, including M=0, still yields a CVaR evaluation gap for the policy returned by CVaR VI. This suggests the affected MDPs are actually not isolated and the CVaR DP evaluation fails in general for the CVaR VI policy. This is not to say however that *all* risk-dependent policies suffer from a CVaR evaluation gap and may well be, in fact, a consequence of CVaR VI optimizing over a too permissive set of risk-dependent policies.
>
> For instance, a policy $\tilde{\pi}(s_1,y)$  that selects $a_1$ if $y \geq 0.25$ and $a_2$ instead will not suffer from a CVaR evaluation gap at $\alpha=0.5$. This particular policy would however in turn have CVaR evaluation gap at other values of $\alpha$, including $\alpha=0.25$ for example.
>
> In summary, examples showing the presence and the absence of CVaR evaluation gaps abound both ways and, as we suggest in our conclusion, an interesting and open question remains to understand how can these gaps be eliminated for good, e.g. by using a different algorithm than CVaR VI or by imposing restrictions on MDPs and policies.

---

### Author Response · Authors · 2026-01-23
**PDF Update**

We have updated the PDF to now include our changes made following initial reviewer comments.

---

### Decision · Action_Editor_8ozW · 2026-03-09

**Recommendation:** Reject

**Additional Comments:**

The paper can make a valuable contribution, but the clarity and comprehensibility of the theoretical contribution and the proofs have not yet reached the required quality. The paper should be revised accordingly, and the comments of Reviewer KU88 should be taken into account point by point.

**Audience:**

Yes

**Audience Explanation:**

All reviewers indicate that the paper is of interest to at least a subset of TMLR readers. In particular, the work addresses an ML-theory–relevant question in risk-sensitive DP/RL by clarifying when static dual CVaR DP decompositions can yield sub-optimal, risk-dependent policies, which makes the findings relevant for researchers working on risk measures and principled decision-making methods.

**Claims And Evidence:**

No

**Claims Explanation:**

Even after the rebuttal “Claims & Evidence” is not uniformly considered to have been fulfilled. In particular, the responses to the points of criticism raised by Reviewer KU88 are deemed to be not clear enough.

The following reviewer points are not clearly / explicitly addressed in the Authors’ response (or are only addressed in a way that doesn’t directly answer the reviewer’s specific question):

1.	Lemma 1, Q1 (“How is the min defined… What is the nature of the space / structure?”)
The Authors say they “explicitly define” the term before Definition 2, but they don’t explicitly confirm they clarified the underlying space/structure over which the minimum is taken (e.g., what set the minimization ranges over, what constraints define it, what mathematical object it is).
2.	Lemma 1, Q2 (“min is monotonous… monotonous with respect to what?”)
The response says they reworked the proof and made steps explicit (upper bound via perturbations, cancellation of identical terms, existence of some ), but it still doesn’t directly answer Reviewer KU88’s pinpoint question: which variable/argument is the monotonicity in (and under what ordering/conditions).
3.	Writing/conceptual clarity, Point 2 (Figure 3: ask for “counterintuitive insight or something meaningful … after the equation”)
They add an interpretive sentence (“risk-seeking picks , risk-averse picks ”), which is helpful, but it doesn’t explicitly deliver what Reviewer KU88 asked for: a clear ‘insight’ takeaway (possibly counterintuitive, or at least a highlighted implication) tied to the equation/figure beyond describing what it shows.

**Resubmission Of Major Revision:**

The authors may consider submitting a major revision at a later time.